# The Effectiveness of Digital Interventions for Psychological Well-Being in the Workplace: A Systematic Review Protocol

**DOI:** 10.3390/ijerph17010255

**Published:** 2019-12-30

**Authors:** Maria Armaou, Stathis Konstantinidis, Holly Blake

**Affiliations:** 1School of Health Sciences, University of Nottingham, Nottingham NG7 2HA, UK; stathis.konstantinidis@nottingham.ac.uk (S.K.); holly.blake@nottingham.ac.uk (H.B.); 2National Institute for Health Research (NIHR) Nottingham Biomedical Research Centre, Nottingham, NG7 2UH, UK

**Keywords:** systematic review, psychological well-being, digital interventions, workplace

## Abstract

Objective: Psychological well-being has been associated with desirable individual and organisational outcomes. This systematic review aims to assess the effectiveness of digital interventions for the improvement of psychological well-being and/or the prevention/management of poor mental well-being in the workplace. Methods: This review protocol is registered in PROSPERO (CRD42019142428). Scientific databases including MEDLINE, Web of Science, CINAHL, PsycINFO, Cochrane Register of Controlled Trials (CENTRAL) and EMBASE will be searched for relevant studies published between January 1990 and July 2019. Studies will be included if they report specific primary and secondary outcomes of digital interventions delivered to adults in the workplace for the improvement of their psychological wellbeing and/or the prevention/management of poor mental well-being and were published in English. Following screening of titles and abstracts, full texts of potentially eligible papers will be screened in duplicate to identify studies that assess the effectiveness of those digital interventions. Discrepancies will be resolved through consensus or by consulting a third reviewer. An integrated narrative synthesis will assess included studies’ findings, and a meta-analysis will be performed if included studies appear to be homogeneous. The “Cochrane Collaboration’s Risk of Bias” tool and the JBI (Joanna Briggs Institute) Critical Appraisal Checklist for Quasi-Experimental Studies will be used to appraise included studies. Conclusion: The results of this work will provide recommendations on the use of digital interventions for the promotion of psychological well-being at work. It will also guide the development of future workplace digital interventions and subsequent primary research in this field.

## 1. Introduction

Digital interventions are becoming increasingly popular for the improvement of mental well-being in organisations due to their cost-effectiveness, their potential to reach a larger number of individuals, and their ability to offer greater anonymity and stigma reduction compared to traditional face-to-face group-based interventions [1,2]. There is substantial evidence on the impact of poor mental well-being in the workplace on individuals and organisations, which explains the need for the development and evaluation of such interventions. Common mental health problems such as depression, anxiety and stress are associated with reduced productivity, while increased sickness absence, presenteeism and poor mental health in UK is estimated to have an annual cost between £33–42 billion [3]). In 2015, 12% of additional temporary National Health Service (NHS) staffing was needed due to urgent staff shortages due to absenteeism [4]), and according to the latest NHS staff survey, 40% of the staff attributed feeling unwell due to work-related stress [5]). For this reason, the improvement of employees’ poor mental well-being has been the focus of stress prevention and stress management interventions, as well as the focus of mental health promotion interventions, and there is substantial evidence suggesting their effectiveness [6,7,8,9,10,11]). At the same time, there is long-established evidence of the importance of psychological well-being (PSW) for desirable individual and organisational outcomes [12,13,14]).

Stress prevention largely refers to the modification of risk factors (e.g., job stressors) that can undermine employees’ well-being, while stress management refers to the improvement of individuals’ coping skills and overall stress management skills before they develop symptoms. For example, stress management can involve individual-level stress management interventions that aim to alleviate the symptoms of stress in the workplace by supporting the development of diverse coping skills such as cognitive–behavioural skills training, relaxation techniques training, exercise and goal setting [8,15]. At times, however, the distinction between stress prevention and stress management interventions may be less clear. For example, organisational-wide skills training courses that aim to empower all employees—and not only those at “risk”—to improve their stress management capabilities can be also viewed as a stress prevention strategy [16]. Furthermore, secondary prevention is increasingly viewed as part of organisations’ well-being programmes [17,18] integrated within organisations’ mental health promotion strategies. For this reason, the theoretical underpinning of stress management interventions and their evaluation is also echoed in guidelines pertaining to the prevention of mental health problems at work [11,19,20]. 

The concept of psychological well-being goes beyond the absence of mental health problems and symptoms and can address psychological parameters such as subjective well-being, autonomy, positive relationships, sense of purpose and personal growth [21]. Applied research has viewed PSW as a subjective experience, as a state on which positive emotions relatively surpass negative emotions, and as a form of global judgement on life domains [22]. It has been consistently linked to health and mental health outcomes for individuals, while on the organisational level, psychological well-being has been associated with performance indicators and a range of organisational outcomes (e.g., turnover rates, absenteeism, employee engagement) [23]. For example, subjective indicators of well-being can effectively indicate the incidence of short-term absences [24]. Similarly, there is growing attention to the premise of psychological interventions for psychological well-being through the development of positive cognitive/affective states (e.g., work engagement, resilience, optimism) and their importance in creating resilient workplaces and engaged workforce [25,26,27,28,29,30]. Psychological well-being promotion in organisations has been associated with the development and interplay of resources at the individual-, group-, leader-, and organisational levels [31,32,33]. Resources can be generically defined as “anything perceived by the individual to help attain his or her goals” [34] (p. 6) and Nielsen et al.’s [32] meta-analysis, drawing from current developments in occupational health psychology and Human Resource Management (HRM) perspectives, supports a four-level model of resources describing psychologically healthy organisations. Their model describes the positive impact of personal resources (e.g., positive psychological capital, self-efficacy, resilience), group resources (e.g., social support), leader resources (e.g., transformation leadership) and organisational resources (e.g., autonomy and mutually beneficial HRM practices) on employees’ well-being and performance [31,32,33]. For this reason, psychological well-being promotion in the workplace is largely dependent on the focus of the interventions, how their effectiveness is measured and how they may be integrated within individuals’ organisational contexts, and their wider sociocultural environments. For example, studies may assess the effectiveness of psychological wellbeing promotion interventions within or across different occupational settings, intervention types (e.g., Cognitive Behavioural Therapy (CBT)-based, mindfulness-based, stress management), outcomes (e.g., cognitive, affective and behavioural responses) or population characteristics (e.g., targeting “at risk” employees or all employees universally) [10,35,36]. However, due to the nature of those interventions, randomisation most frequently refers to the allocation of participants in different types of interventions or a control condition.

Although there is growing evidence on the premise of digital interventions in the workplace, issues pertaining to their theoretical basis and their methods have posed a considerable challenge in the systematic evaluation of their effectiveness. Similar to non-digital interventions, they can offer primary or secondary prevention strategies and self-management directions, but findings from previous reviews cannot be generalised to the rapidly growing palette of digital workplace interventions. Previous systematic literature reviews have found that that group-based universal and targeted workplace interventions, whose contents incorporate Cognitive Behavioural Therapy approaches, can be effective in reducing anxiety and depression and in the workplace [9,35]. However, generalisations elicited at the group level may not account for individual variations over time or be compatible with the context of digital interventions [37,38]. For example, it has been suggested that digital interventions need to be theoretically based on dynamic psychological models and theories of behaviours, and have features that can be responsive to large quantities of real-time data [37,38]. Furthermore, diverse modes of delivery and difficulties in setting comparators or a testing environment can pose further challenges to the theoretical basis of digital interventions and the evaluation of their effectiveness [37]. For example, there is some evidence for the effectiveness of stress management interventions in nurses either as part of larger eMental health programmes or as standalone eHealth interventions or eHealth modules [22,39,40,41]. This is a characteristic example of the theoretical disparity of digital stress management interventions [42]. Although evidence supports the use of theory-informed health behaviour interventions [43], further evaluation and refinement is required to strengthen the evidence base of effective interventions across different groups and in different contexts [44,45]. For this reason, there is a need to review the evidence pertaining to the effectiveness of stress management interventions that are delivered via digital technologies.

Previous systematic literature reviews on the effectiveness of digital interventions towards preventing or managing poor mental well-being in the workplace have focused on the reduction of stress and poor mental health symptoms without addressing associations with other psychological, behavioural or occupational outcomes. For example, Stratton et al.’s [36] review examined the effectiveness of eHealth interventions in reducing mental health conditions in the workplace, demonstrating that stress management interventions had the highest levels of heterogeneity and small effect sizes compared to CBT and mindfulness interventions, highlighting their differential effects when administered to targeted employees versus universally to all of the employees. Furthermore, Carolan et al. [14] meta-analysis concluded that digital mental health interventions can be effective in improving employees’ psychological wellbeing, measured as stress and/or depression reduction and work effectiveness, highlighting the importance of addressing associations with a wider range of occupational outcomes. At the same time, there is significantly less rigorous research in examining the effectiveness of digital interventions for the improvement of psychological well-being exploring the links with other psychological and organisational outcomes [16,46]. This systematic review aims to address this gap in the literature by (a) identifying studies that report the effectiveness of digital interventions for the promotion of psychological well-being in the workplace, (b) assessing the relationship between intervention effectiveness and its theoretical underpinnings, and (c) allowing an exploration of their associations with occupational outcomes.

### 1.1. Review Aim

The aim of this systematic literature review is to assess the effectiveness of digital interventions designed to improve the psychological well-being and/or the prevention/management of poor mental well-being in the workplace.

### 1.2. Review Objectives

(1)To describe measures that have been utilised to assess the effectiveness of digital interventions designed to improve the following:
(i)Psychological well-being at work;(ii)The prevention/management of poor mental well-being in the workplace.
(2)To identify associations of digital interventions designed to improve the psychological well-being and/or the prevention/management of poor mental well-being in the workplace with individual and organisational outcomes.(3)To assess whether theory-informed digital interventions for the promotion of psychological well-being are more effective or not, and
(i)to identify which theory or mechanism is associated with greater effectiveness.


## 2. Methods

The protocol for this systematic review was developed using the Preferred Reporting Items for Systematic Review and Meta-Analysis (PRISMA) guidelines [47] (Appendix A) and has been registered with the PROSPERO International Prospective Register of Systematic Reviews (registration number: CRD42019142428). Following Cochrane guidance on the development of a systematic review protocol, the research questions formed the base of a search strategy that was further developed using PICO-elements (Population, Interventions, Comparators, Outcomes). Studies will be assessed against clearly defined criteria to determine their inclusion or exclusion in the review; the findings of included studies will be assessed and reported [48]. 

### 2.1. Eligibility

#### 2.1.1. Inclusion Criteria

a)Type of participants:The included studies will report the result of interventions targeting “employees” defined as working-age adults, as well as adults over 65 years that are still in a contracted role within their organisations.b)Context/Setting:Included studies will report interventions delivered in the workplace setting.c)Type of Interventions:Included studies will report interventions that are delivered using digital technology via any delivery channel (e.g., these will include but are not limited to web-based interventions, email, mobile phone and apps). Interventions will focus on psychological or mental well-being (e.g., behavioural, cognitive or educational interventions) delivered via any type of digital technology via any delivery channel. Both controlled (e.g., reporting comparisons with a control group, another intervention, face-to-face intervention) and uncontrolled (e.g., single-group studies, pilot and feasibility studies) studies will be considered for inclusion. There will be no restrictions with regards to the timing, duration, or modality of the interventions.d)Type of studies:The types of the studies that will be included in this review will be experimental (e.g., randomised controlled trials, cluster-randomised Controlled Trials (RCTs), controlled before-and-after studies) or quasi-experimental studies (e.g., one-group pre-test-post-test design, time series designs). Furthermore, included studies will provide an analysis of the results of the intervention.e)Comparator(s)/control:Included studies will report digital interventions that are compared with any other type of interventions or a no intervention control group, or have no comparator group. Controlled-studies (e.g., reporting comparisons with a no intervention control group, alternative form of digital intervention or a non-digital intervention) and uncontrolled studies (e.g., single group pre-test post-test studies, uncontrolled pilot and feasibility studies) will be considered for inclusion. Separate analysis will be presented for those studies with randomisation to comparator/control groups.f)Type of publication:Included studies will report empirical research published in peer-reviewed journals or conference proceedings that are accompanied with full-length peer-reviewed papers.g)Outcome measures:Included studies will report interventions for which the primary outcome is a focus on the improvement of psychological well-being and/or prevention/management of poor mental well-being.

The primary goal of all included interventions is to improve psychological well-being and/or improve psychological indicator(s). For this reason, included studies will need to report on at least one instrument that claims to measure psychological well-being and/or mental well-being outcome(s). Primary outcomes may include measures of (a) common mental well-being outcomes at work (e.g., perceived anxiety, depression), (b) work-related well-being measures (e.g., perceived stress, burnout, work-engagement,) and/or (c) measures of psychological indicators for mental well-being at work (e.g., positive psychological capital, resilience, self-efficacy, coping strategies, optimism).

Secondary outcomes may include measures of physical health, other psychosocial outcomes (e.g., social support), behavioural outcomes (e.g., changes in, or intentions to change, health behaviours), or organisational outcomes (e.g., return to work, sickness absence, productivity, performance, turnover etc). This will allow for the assessment of any associations between psychological well-being and/or mental well-being outcomes with other individual-level outcomes as well as outcomes of importance to employers. 

#### 2.1.2. Exclusion Criteria

a)Type of participants:The study will exclude children and young people (under 18 years of age), and retirees.b)Context/setting:Studies will be excluded if they report interventions delivered in settings other than the participants’ workplace. c)Type of interventions:Studies will be excluded if (i) they report digital interventions delivered in conjunction with other interventions (since it would be difficult to ascertain the unique contribution of the digital element), and/or (ii) they do not include a psychological intervention (such as reporting on digital interventions without a psychological component that primarily target physical/behavioural outcomes; i.e., weight loss, physical activity, tracking alcohol consumption).d)Type of studies:Studies will be excluded if they are not original intervention studies, if they are published in journals that are not peer-reviewed, or if they are not in the English language. They will be excluded if they do not report on a digital intervention. Previous relevant systematic reviews will be identified for the purpose of identification of primary studies, but they will not be included in the review. Studies that report case studies, cohort studies, cross-sectional research designs, conference abstracts (without a corresponding full-length peer-reviewed paper) and unpublished research (e.g., unpublished dissertations/theses) will not be included.e)Comparator(s)/control:Studies will be excluded if they do not use a randomised or quasi-experimental design (e.g., the review will include randomised and non-randomised studies with comparator/control groups as well as those using a one-group pre-test- post-test design, or time series design).f)Outcome measures:Studies will be excluded if they (i) do not include relevant outcome measures, (ii) report on interventions or outcomes that focus primarily on the clinical treatment of mental health disorders (e.g., Post-traumatic stress disorder (PTSD), major depression), and (iii) if their primary outcomes do not measure psychological well-being and/or mental well-being outcome(s).

### 2.2. Search Strategy

A comprehensive literature search will be conducted using electronic databases including MEDLINE, Web of Science, CINAHL, PsycINFO, Cochrane Register of Controlled Trials (CENTRAL) and EMBASE to identify records that match our inclusion criteria published from January 1990 to till July 2019. The search strategy was pilot tested in PsychInfo and has been refined and appropriately modified for use with each database (Appendix A). The search main terms were “psychological well-being”, “mental wellbeing”, “digital interventions” and “workplace”. Search terms will be combined with the appropriate Boolean operators (“OR”, “AND” and “NOT”) and/or thesaurus terms. Terms will be searched in titles, abstracts and keywords. A hand search of the reference lists of included articles and related systematic reviews will be checked in order to identify additional potentially eligible studies.

### 2.3. Selection Processes

Two reviewers will participate in the selection process. A Mendeley desktop will be used to store references and subsequently identify and remove duplicates. All titles and abstracts will be screened for eligibility by one reviewer, and for those that remain unclear, full texts will be sought. Abstracts and full texts of potentially eligible studies will then be screened independently by two reviewers against the studies’ eligibility criteria, taking into account the intervention type, study population, and the reported outcomes. The two reviewers will reach agreement through discussion, and if consensus is not reached, a third reviewer will provide resolution. Reasons for exclusion will be documented at each stage of the selection process, which will be visualised with a flow diagram following the PRISMA guidelines.

### 2.4. Data Extraction

Data extraction will be performed independently by two reviewers, and consensus will be reached through discussion. The JBI data extraction form for experimental/observational studies [49] will be used in order to extract all relevant information from the study. In particular, the form captures information about the study method (e.g., RCT, quasi-RCT, longitudinal), study participants and interventions (setting, population, sample size, intervention/comparator details), outcome measures (outcome description and measures) and results (outcome data for dichotomous and continuous variables).

### 2.5. Quality Appraisal

Two reviewers will independently conduct the quality appraisal of the retrieved papers using the Cochrane Collaboration’s Risk of Bias and the JBI Critical Appraisal Checklist for Quasi-Experimental Studies. The “Cochrane Collaboration’s Risk of Bias” tool [50] will be used to measure the risk of bias, with items judged as low, high or unclear risk for the included studies that adopt a randomised design. Bias is assessed as a judgment (high, low, or unclear) for individual studies from five domains including selection bias (random sequence generation, allocation concealment), performance bias (blinding of participants and personnel), detection bias (blinding of outcome assessment), attrition bias (incomplete outcome data), reporting bias (selective reporting), and other (any important concerns about bias not already discussed).

The JBI Critical Appraisal Checklist for Quasi-Experimental Studies (non-randomised experimental studies) [51] will be used to evaluate the quality of the included quasi-experimental studies. Bias is assessed as a judgement (Yes, No, Unclear, n/a) through nine questions that aim to evaluate the study’s research design and the validity of its results. In particular, the checklist assesses the study’s casual relationships, the similarities of people in the compared groups and the type of care they received, the existence of a control group, the existence of pre- and post-intervention measurements, the procedures of any follow-up measures, the measure of the outcomes included in any comparisons, the reliability of the outcomes, and the appropriateness of the statistical analysis.

### 2.6. Data Synthesis

It is expected that the included studies will be highly heterogeneous in their design and outcome measures. An integrated narrative synthesis will be performed in order to explore the relationships between the findings within and across the included studies as outlined by Popay et al. [52]. This is a six-step iterative process including (1) developing a theory, (2) preliminary analysis, (3) the exploration of relationships, (4) assessment of robustness, and (5) conclusions and recommendations. Specific tools and techniques will be used to facilitate this knowledge synthesis process (content analysis and tabulation, concept mapping and critical reflection) [52,53,54] (Figure 1). The preliminary synthesis through the content analysis and tabulation process will produce textual descriptions of all studies and construct a common rubric organising studies into logical categories. Then, the exploration of the relationships within and between the studies through concept mapping and a visual representation of the relationships within and between studies will be used to increase conceptual and methodological triangulation. Finally, the assessment of the robustness of the synthesis through critical reflection and quality appraisal will allow the construction of a best evidence synthesis and inform the review’s conclusions and recommendations, elaborating on the factors that can explain the effectiveness of digital interventions for psychological well-being in the workplace.

An advantage of this approach is that it can allow an investigation of the heterogeneity of the included studies and highlight variations that may be attributable to theoretical constructs. Such an analysis will also allow us to explore potential relationships between the theoretical underpinnings of different interventions and their effectiveness. In particular, it will allow a complete description of the constructs, measures, intervention characteristics and research designs that researchers utilised to improve psychological well-being at work (Table 1). Its process will allow findings to be presented in a narrative form including tables and figures to aid in data presentation focusing on three types of primary outcomes: (a) common mental health problems at work, (b) work-related well-being, and (c) psychological indicators for mental well-being at work.

Those studies with randomisation to a comparator/control group will be separately analysed, and if they appear to be adequately homogeneous in terms of design and comparator, only then will a meta-analytic (quantitative) pooling be performed using Review Manager (RevMan) Version 5.3. [55]. 

If the variability, however, is too high, then only the above narrative synthesis will be performed.

Summary measures: The included studies will report individual-level interventions with primary individual-level outcomes. For each study, between-group effect sizes will be computed using Cohen’s d. When necessary, standard deviations will be reconstructed from *p*-values or t-statistics.

Analysis: We will use a random-effects model and a 95% confidence interval with two-tailed tests.

Effect size interpretations: Effect sizes will be reported either as odds ratios (for dichotomous data) or weighted (or standardized) mean differences (for continuous data), and their 95% confidence intervals will be calculated for analysis. Effect sizes will show the changes reported from baseline standardised measures of psychological resources, poor mental well-being outcomes (e.g., depression, perceived anxiety) and work-related well-being, (e.g., perceived stress, burnout, work-engagement). When available, between-group effect sizes (Cohen’s d) will be computed for follow-up differences in outcome measures.

Heterogeneity: Heterogeneity will be assessed statistically using the standard chi-squared x^2^, I^2^ and the Q_within_ statistic. A significant Q_within_ value would reject the null hypothesis of homogeneity, while the I^2^ statistic shows the proportion of observed variance that is not due to sampling error and indicates heterogeneity in percentages (ranging from no heterogeneity to high levels of heterogeneity) [56]. 

Small study effect: Funnel plots will be used to assess the small study effect for the overall effects. Adjusted effect sizes will be calculated for funnel plots demonstrating significant asymmetry (calculated as Egger’s intercept) and the Duval and Tweedie trim and fill will be used to quantify the magnitude of publication bias [57].

## 3. Practical Policy Implications

The present study will offer an assessment of the effectiveness of individual-level interventions for the promotion of psychological well-being in the workplace. This knowledge will be valuable for researchers and professionals seeking to design such interventions, as it will offer the following:
Recommendations on the use of digital interventions, their strengths and limitations;Guidance in identifying specific indicators of psychological well-being in the workplace that can be targeted via an individual-level intervention;An understanding of the theories that guide such interventions and the role of theory in their development;An exploration of a range of individual-level outcomes, assessed via standardised measures, and their associations with other individual and organisational outcomes.

## 4. Conclusions

The effectiveness and limitations of individual-level interventions for stress management and mental health promotion in organisations is well-documented. However, the published evidence regarding the effectiveness of digital interventions for employees’ psychological well-being is still lacking, which has often associated with theoretical and methodological differences across these types of interventions. This systematic review will be the first to assess the effectiveness of psychological interventions delivered via digital technologies in the workplace for employees’ psychological well-being (i.e., stress reduction and/or the improvement of psychological indicators for the prevention/management of poor mental well-being) and to explore their associations with organisational and behavioural outcomes (e.g., return to work, health behaviour change). The review will provide valuable insight into the range of digital interventions that are available to employees and their effectiveness in improving their psychological well-being. Its findings can offer guidance for the development and implementation of future digital interventions aiming to improve mental well-being in organisations.

## Figures and Tables

**Figure 1 ijerph-17-00255-f001:**
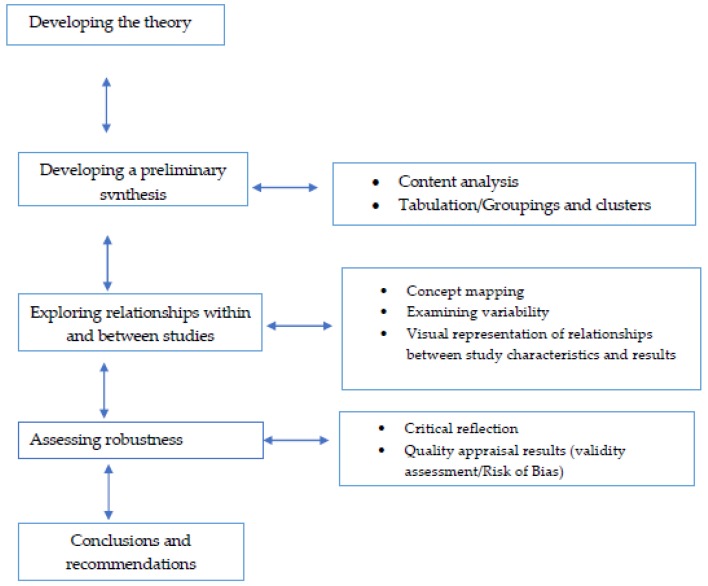
Integrative narrative synthesis process (Adapted from Popay J, Roberts H, Sowden A, Petticrew M, Arai L, Rodgers M, Britten N, Roen K, Duffy S. [52]. Guidance on the Conduct of Narrative Synthesis in Systematic Reviews. A product from the Economic and Social Research Council (ESRC) Methods Programme.

**Table 1 ijerph-17-00255-t001:** Description of narrative synthesis.

Psychological Well-Being		
Interventions targeting common mental well-being outcomes at work (e.g., perceived anxiety, depression),	Interventions targeting work-related well-being (e.g., perceived work stress, burnout, work-engagement)	Interventions targeting psychological indicators for mental well-being at work (e.g., positive psychological capital, resilience, self-efficacy, coping strategies, optimism).
Concepts and constructs for psychological well-being at work		
Description of characteristics of interventions		
Description of measures and outcomes (primary outcomes and associations with secondary outcomes		
Research designs		

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
