# Peer review of "The Effectiveness of Digital Interventions for Psychological Well-Being in the Workplace: A Systematic Review Protocol"

_ijerph, 2019, doi:10.3390/ijerph17010255_

Round 1

Reviewer 1 Report

This is a well written outline of a prospective systematic review which sets out the aims of the review clearly and will contribute to the current knowledge about the effectiveness of digital interventions in the workplace. However, I think some minor changes can improve the quality of the manuscript:

Some grammatical errors will need to be corrected in the following sentence (highlighted in bold)

(p. 2, line 89- 92) "However, generalisations elicited on at group-level may not account for individual variations over time or be compatible to with/ in (depending on the meaning of this sentence?) the context of digital interventions, while the diversity in their delivery modalities and interventions’ content poses further challenges to their theoretical basis and the evaluation of their effectiveness (Michie et al., 2017; Hekler et al., 2016)"

In general I found this sentence syntactically and conceptually convoluted and this complex structure is making the sentence unclear, so it may better to re-phrase and unpack the concepts and the ideas a bit more.

Another minor grammatical error to be corrected (p.7, lines 301-302): "Effect sizes will be reported either as odds ratios (for dichotomous data) and or weighted (or standardized) mean differences (for continuous data) and their 95% confidence intervals…"  More importantly, I believe the review should assess whether theory-informed interventions are more effective (or not) and which theory or mechanism of action is linked to higher effectiveness (perhaps to explicitly state this as a 3rd objective?) given that the review is looking to evaluate the evidence about theory-informed health interventions being associated with greater effectiveness (p3, lines 97-99).

Overall, this is a well-described and potentially very useful review in this area.

Reviewer 2 Report

Comments 1. Is the use of two reviewers in the selection process a standard of the literature or not? 2. There is an issue regarding the interpretation of the estimates. Employees are not randomly assigned into workplaces. Failure to account for sorting of employees will bias any estimated effects. The size of the bias is not known. This problem can be addressed using information on employees’ wage and work histories (https://doi.org/10.1016/j.jebo.2012.09.005). This issue should be noted in the revised version. 3. The paper does not pay attention to the potential heterogeneity in the estimated effects. The relationships can differ significantly e.g. by gender and/or age. 4. The concluding section of the paper should provide practical policy implications that stem from the results.
